# Age, Growth, and Natural Mortality of Graysby, *Cephalophilis cruentata*, from the Southeastern United States

**Michael L. Burton \*, Jennifer C. Potts, Andrew D. Ostrowski and Kyle W. Shertzer** 

National Oceanic and Atmospheric Administration, National Marine Fisheries Service, Southeast Fisheries Science Center, Beaufort Laboratory, 101 Pivers Island Road, Beaufort, NC 28516, USA
\* Correspondence: Michael.Burton@noaa.gov; Tel.: +1-252-728-8756

**Abstract:** Graysby (*Cephalophilis cruentata*) ($n$ = 1308) collected from the southeastern United States Atlantic coast from 2001 to 2016 were aged using sectioned sagittal otoliths. Opaque zones formed February to June (peaking in April). Ages ranged from 2 to 21 years, and the largest fish measured 453 mm TL. Growth morph analysis revealed two regionally distinct growth trajectories: von Bertalanffy growth equations were $L_t = 388\ (1 - e^{-0.12(t+5.73)})$ for fish from North Carolina through southeast Florida (northern region), and $L_t = 267\ (1 - e^{-0.17(t+6.20)})$ for fish from the Florida Keys (southern region). When growth was re-estimated using a fixed $t_0$ value of −0.75 to estimate for smaller fish, growth equations were $L_t = 349\ (1 - e^{-0.26(t+0.75)})$ and $L_t = 250\ (1 - e^{-0.43(t+0.75)})$ for fish from the northern and southern regions, respectively. The age-invariant estimate of natural mortality was $M$ = 0.30 for all fish, while age-specific estimates ranged 0.88–0.28 $y^{-1}$ for fish aged 1–21 from the northern region and 0.89–0.47 $y^{-1}$ for fish aged 1–15 from the southern region. This study presents the first comprehensive analysis of life-history parameters for graysby from the Atlantic waters off the southeastern United States, including specimens from both recreational and commercial fisheries.

**Keywords:** ageing; graysby; growth morphs; natural mortality

## 1. Introduction

Graysby (*Cephalophilis cruentata* Lacepède 1802) (Family Serranidae) is a small member of the grouper family in the tropical western Atlantic, infrequently attaining lengths greater than 400 mm (mm) total length (TL). The species is a protogynous hermaphrodite, changing from female to male during its lifetime. Their distribution ranges from Brazil northward to North Carolina and Bermuda [1] and is abundant throughout the Caribbean, being the most commonly observed serranid in coral reef habitats off La Parguera, Puerto Rico [2]. Adults typically inhabit subtropical and tropical rocky ledge and coral reef areas, but adult graysby have been collected in Jamaican seagrass beds of 2–4 m depth [3]. They were commonly observed at depths up to 170 m in Jamaica and 145 m in Belize [4].

Graysby are currently included in the South Atlantic Fishery Management Council's Snapper-Grouper Fishery Management Plan [5]. The stock is regulated by inclusion in an aggregate grouper bag limit of three fish per person per day in the recreational fishery, and by inclusion in a spawning season closure of the shallow water grouper complex from January 1 through April 30 of each year [the complex includes gag *Mycteroperca microlepis* (Goode and Beane 1879); black grouper *Mycteroperca bonaci* (Poey 1860); red grouper *Epinephelus morio* (Valenciennes 1828); scamp *Mycteroperca phenax* (Jordan and Swain 1884); rock hind *Epinephelus adscensionis* (Osbeck 1765); red hind *E. guttatus* (Linnaeus 1758); coney *Cephalophilis fulva* (Linnaeus 1758); yellowfin grouper *Mycteroperca venenosa* (Linnaeus 1758); and yellowmouth grouper *M. interstitialis* (Poey 1860)]. The stock is also

included in the annual catch limits (ACL), or quotas, for the shallow water grouper complex. The ACLs are currently set at 25,193 kg for the commercial sector and 22,066 kg for the recreational sector. There are currently no size limits on graysby in either fishery sector.

Graysby are of limited economic importance to the southeastern United States (SEUS, North Carolina to Florida Keys, including the Dry Tortugas) reef fish fishery. Estimated annual landings of graysby from headboats (vessels carrying at least seven anglers engaged in recreational fishing) sampled by the Southeast Region Headboat Survey (SRHS) averaged 2982 fish (1503 kg) during the period 1981–2017 [6] Estimated annual landings by anglers fishing from private recreational boats and charter boats averaged 11,772 fish (4118 kg) during the period 1981–2017 (unpub. data, available at http://www. st.nmfs.noaa.gov/recreational-fisheries/data-and-documentation/queries/index). Combined landings from all recreational sectors show no consistent increasing or decreasing trends with the majority of fish coming from Florida-Georgia waters (Figure 1). Total commercial landings of graysby in the SEUS during the period 1991–2015 were 34,986 kg, with 96% of these coming from FL-GA waters (unpub. data, available at: https://www.st.nmfs.noaa.gov/st1/commercial/landings/annual_landings.html).

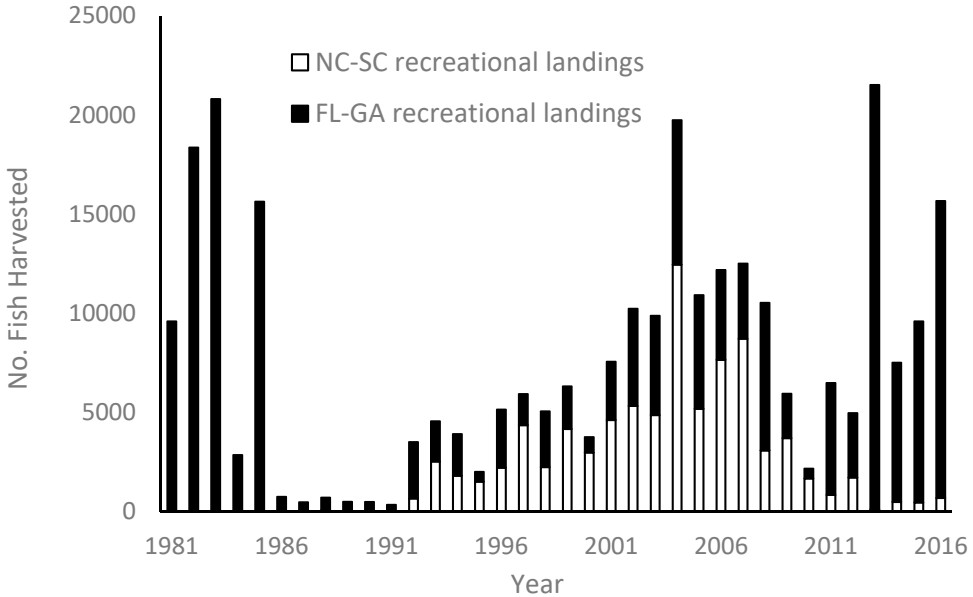

**Figure 1.** Combined recreational landings of graysby (*Cephalophilis cruentata*) from the southeastern United States, 1981–2016.

We studied graysby in the SEUS because of the increasing need for stock assessments of data-limited species. Our analyses relied on archived sagittal otoliths collected by long-term, systematic dockside sampling programs. The only previous study of graysby life history in SEUS waters was limited in scope (headboat-caught fish only) and sample size (*n* = 118) [7]. Thus, resource managers have had a paucity of biological information available to use in setting ACLs, the mechanism by which all fish species are now managed under the Magnuson-Stevens Fishery Management and Conservation Act. Our primary goal is to provide updated and comprehensive information on age-growth parameters and natural mortality rates for graysby from the SEUS, filling an important gap for this data-limited species.

## 2. Results

### 2.1. Age Determination and Timing of Opaque Zone Formation

A total of 1318 sagittae from graysby were sectioned. Opaque zones were counted on 1308 (99%) of graysby sections; ten sections were determined to be illegible and excluded from analyses. The majority of samples came from the North Carolina commercial fishery (*n* = 534; 43%) and the Florida Keys recreational fishery (*n* = 294; 22%), respectively. Samples were evenly distributed between fishery

sectors, with 51% of graysby sampled from commercial fisheries and 49% from the recreational sector (Table 1). All fish were measured to the nearest mm TL and, if available, whole weight (g).

**Table 1.** Distribution of otolith samples used in age-growth study of graysby (*Cephalophilis cruentata*) from the southeastern United States, by year, region and fishery sector. NC = North Carolina; SC = South Carolina; EFL = North and central east coast Florida; FLK = southeast Florida–Dry Tortugas.

| | Commercial | | | | Recreational | | | |
|---|---|---|---|---|---|---|---|---|
| **Year** | **NC** | **SC** | **EFL** | **FLK** | **NC** | **SC** | **EFL** | **FLK** |
| 2001 | | | | | | | 4 | |
| 2002 | | | | | | 2 | 3 | |
| 2003 | 2 | | | | 4 | 1 | 11 | |
| 2004 | 94 | | | | | | 10 | |
| 2005 | 120 | 18 | | | 1 | | 17 | 1 |
| 2006 | 6 | 40 | | | | 13 | 17 | |
| 2007 | 33 | 8 | | | 1 | 4 | 12 | |
| 2008 | 50 | 9 | | | | 5 | 1 | 2 |
| 2009 | 42 | 11 | | | | 20 | 13 | 1 |
| 2010 | 21 | | | | 2 | 7 | 7 | |
| 2011 | 13 | 1 | | | 1 | 6 | 4 | 7 |
| 2012 | 51 | 1 | | | 1 | 1 | 13 | 47 |
| 2013 | 27 | 10 | 1 | | 3 | 10 | 20 | 39 |
| 2014 | 31 | 10 | 1 | | 6 | 4 | 29 | 96 |
| 2015 | 20 | 13 | | | 8 | 2 | 30 | 45 |
| 2016 | 24 | 15 | | | 2 | 10 | 37 | 56 |
| Total | 534 | 136 | 2 | | 29 | 85 | 228 | 294 |

Opaque zones on graysby otoliths were moderately easy to interpret (Figure 2), with a within-reader index of average percent error (IAPE) of 3.18% (*n* = 648, or 50% of sections), satisfying the acceptable value for IAPE (5% for species of moderate longevity and reading complexity) [8]. Direct agreement between readings was 61%, and this agreement increased to 91% within ± one year. A final age was determined for all samples, and none were excluded from further analyses. When counting the opaque zones, the entire section was taken into consideration, but the most consistent counts were obtained from the dorsal portion and along the sulcal groove. Some otolith sections exhibited multiple banding, identifiable by discontinuous or incomplete orbits around the core of the section. Multiple banding was noted in earlier studies [1,7].

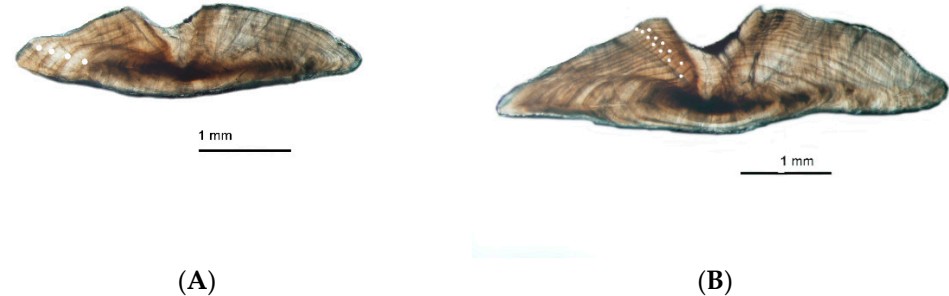

(**A**)　　　　　　　　　　　　　　　　　　　　　　　　　　(**B**)

**Figure 2.** Sections from otoliths of graysby (*Cephalophilis cruentata*); (**A**) 220 mm TL 4-year old; (**B**) 368 mm TL 12-year old.

We were able to assign an edge type to all aged samples for our analysis of opaque zone formation timing. Graysby otoliths exhibited opaque zones on the margin February–June, with a peak in April (Figure 3A). A shift to narrow translucent edge was noted in July, followed by a predominance of

moderate to wide translucent edges from October through January and the widest translucent edges in January and February, immediately prior to opaque zone formation beginning in February (Figure 3B). Graysby otoliths were without an opaque zone on the edge from July through January. We concluded that opaque zones on graysby otoliths were annuli.

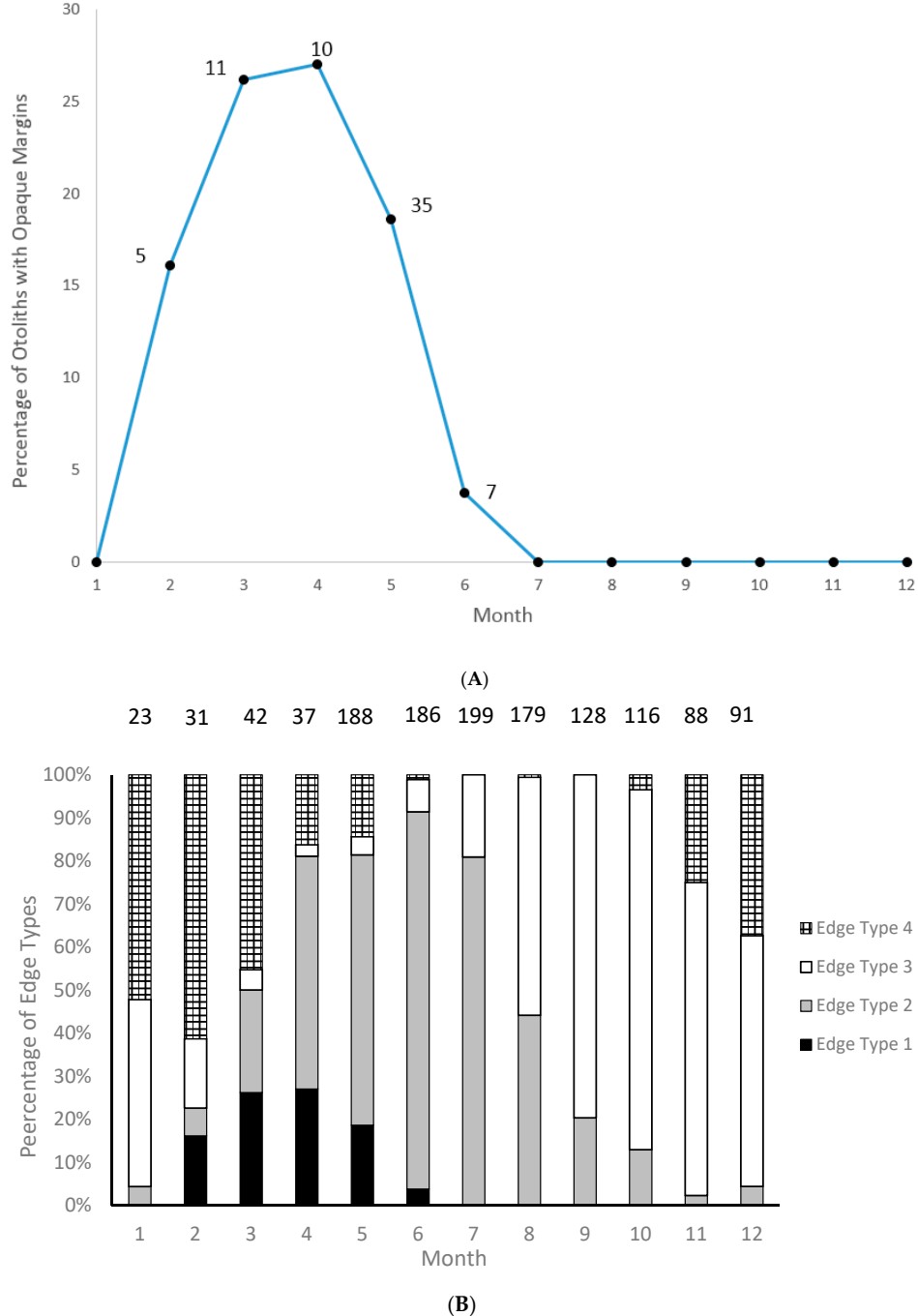

**Figure 3.** Monthly percentages of (**A**) otolith sections with opaque margins on their edges, with monthly sample sizes, and (**B**) monthly percentages of all edge types for graysby (*Cephalophilis cruentata*) collected from the southeastern United States in the period 2001–2016, with total sample sizes above each column. Edge codes: 1 = opaque zone on edge, indicating annulus formation; 2 = small translucent zone, <30% of previous increment; 3 = moderate translucent zone, 30–60% of previous increment; 4 = wide translucent zone, >60% of previous increment [9].

Based on the above-reported timing of opaque zone formation, calendar ages were assigned as follows: for fish caught January through June and having an edge type of 3 or 4, the annuli count was increased by one; for fish caught in that same time period with an edge type of 1 or 2, calendar age was equivalent to annuli count; for fish caught from July to December, the calendar age was equivalent also to the annuli count.

## 2.2. Growth

The initial growth morph analysis with identity treated as unknown provided support for either one or two distinct growth trajectories (Figure A1). Although this initial fitting procedure made no assumptions a priori about the underlying source of each morph, subsequent inspection of the two-morph model showed that the smaller morph comprised fish primarily from the Florida Keys (FLK; southern region) while the larger morph comprised fish from an area encompassing North Carolina through the southeast coast of Florida (NC-EFL; northern region). This observation was supported by statistical analysis of a $2 \times 2$, area-by-morph contingency table ($\chi^2 = 553.9$, df = 1, $P < 0.001$). Size of graysby from the FLK ranged 187–365 mm TL and ages 3–15 (Table 2, Figure 4), while sizes from NC-EFL ranged 185–453 mm TL and ages 2–21 (Table 3, Figure 4). Mean length-at-age was significantly different (paired *t*-tests, $p < 0.05$) between regions for 9 out of 9 ages for which samples were adequate for analysis (Table 4). When the growth morph analysis was repeated using one or two morphs with identity treated as known (FLK or NC-EFL), information criteria provided clear support for the model with two distinct growth trajectories (Figure A2).

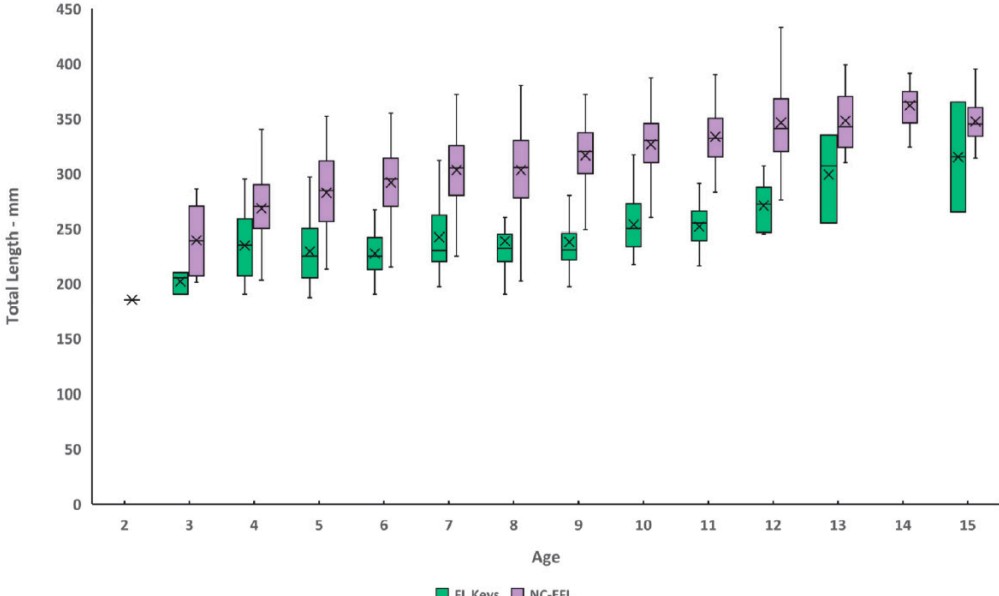

**Figure 4.** Paired mean observed total length at calendar age of graysby (*Cephalophilis cruentata*) caught from North Carolina through southeast Florida (purple) compared to fish caught from the Florida Keys (green). The box and whisker plots indicate mean (x), median (bar), the second through third quartiles (column), and the first and fourth quartiles (whiskers).

**Table 2.** Predicted and observed mean total length (TL, mm) from the von Bertalanffy growth model, natural mortality at age, *M*, and cumulative survivorship to each age for graysby (*Cephalophilis cruentata*) collected during the period 2001–2016 from the Florida Keys based on age-specific estimates of *M* [10] and an age-invariant estimate of 0.30 [11]. Values of age-specific *M* and corresponding survivorship are calculated from sizes at the midpoint of each age (1.5, 2.5, 3.5, etc.). Estimates are based on the model with $t_0 = -0.75$. Standard errors of the means (SE) are provided in parentheses.

| Age | *n* | Mean TL (± SE) | TL Range | Predicted TL | *M* | % Cum Survival Charnov et al. | % Cum Survival Then et al. |
|-----|-----|----------------|----------|--------------|-----|-------------------------------|----------------------------|
| 1 | – | – | – | 156 | 0.88 | – | – |
| 2 | – | – | – | 189 | 0.61 | 41.3 | 74.1 |
| 3 | 3 | 202 (6) | 190–210 | 211 | 0.48 | 22.5 | 54.9 |
| 4 | 32 | 235 (5) | 190–295 | 225 | 0.41 | 14.0 | 40.7 |
| 5 | 25 | 229 (6) | 187–297 | 234 | 0.36 | 9.3 | 30.1 |
| 6 | 37 | 227 (4) | 190–300 | 239 | 0.34 | 6.5 | 22.3 |
| 7 | 42 | 242 (5) | 197–312 | 243 | 0.32 | 4.6 | 16.5 |
| 8 | 35 | 239 (5) | 190–315 | 246 | 0.30 | 3.4 | 12.2 |
| 9 | 46 | 238 (4) | 197–326 | 247 | 0.29 | 2.5 | 9.1 |
| 10 | 33 | 254 (4) | 217–317 | 248 | 0.28 | 1.9 | 6.7 |
| 11 | 23 | 252 (4) | 216–291 | 249 | 0.28 | 1.4 | 5.0 |
| 12 | 13 | 271 (6) | 245–307 | 250 | 0.28 | 1.1 | 3.7 |
| 13 | 3 | 299 (23) | 255–335 | 250 | 0.27 | 0.8 | 2.7 |
| 14 | – | – | – | 250 | 0.27 | 0.6 | 2.0 |
| 15 | 2 | 315 (50) | 265–365 | 250 | 0.27 | 0.5 | 1.5 |

**Table 3.** Predicted and observed mean total length (TL, mm) from the von Bertalanffy growth model, natural mortality at age *M*, and cumulative survivorship to each age for graysby (*Cephalophilis cruentata*) collected during the period 2001–2016 from North Carolina–southeast Florida based on age-specific estimates of *M* [10] and an age-invariant estimate of 0.30 [11]. Values of age-specific *M* and corresponding survivorship are calculated from sizes at the midpoint of each age (1.5, 2.5, 3.5, etc.). Estimates are based on the model with $t_0 = -0.75$. Standard errors of the means (SE) are provided in parentheses.

| Age | *n* | Mean TL (± SE) | TL Range | Predicted TL | *M* | % Cum. Survival Charnov et al. | % Cum. Survival Then et al. |
|-----|-----|----------------|----------|--------------|-----|--------------------------------|-----------------------------|
| 1 | – | – | – | 155 | 0.88 | – | – |
| 2 | 1 | 185 | – | 200 | 0.60 | 41.4 | 74.1 |
| 3 | 10 | 239 (10) | 201–286 | 234 | 0.48 | 22.6 | 54.9 |
| 4 | 81 | 268 (3) | 203–372 | 261 | 0.41 | 14.1 | 40.7 |
| 5 | 96 | 282 (4) | 213–352 | 281 | 0.36 | 9.4 | 30.1 |
| 6 | 143 | 292 (3) | 215–355 | 297 | 0.33 | 6.5 | 22.3 |
| 7 | 138 | 303 (3) | 209–372 | 309 | 0.31 | 4.7 | 16.5 |
| 8 | 116 | 303 (3) | 202–380 | 318 | 0.30 | 3.4 | 12.2 |
| 9 | 95 | 316 (3) | 249–372 | 325 | 0.29 | 2.5 | 9.1 |
| 10 | 74 | 327 (4) | 229–400 | 331 | 0.28 | 1.9 | 6.7 |
| 11 | 79 | 333 (3) | 237–410 | 335 | 0.28 | 1.4 | 5.0 |
| 12 | 52 | 346 (5) | 276–446 | 338 | 0.27 | 1.1 | 3.7 |
| 13 | 38 | 348 (5) | 310–453 | 341 | 0.27 | 0.8 | 2.7 |
| 14 | 21 | 362 (4) | 324–391 | 343 | 0.27 | 0.6 | 2.0 |
| 15 | 31 | 347 (4) | 291–395 | 344 | 0.27 | 0.5 | 1.5 |
| 16 | 15 | 352 (4) | 320–380 | 345 | 0.27 | 0.4 | 1.1 |
| 17 | 11 | 360 (6) | 337–390 | 346 | 0.26 | 0.3 | 0.8 |
| 18 | 10 | 358 (6) | 339–395 | 347 | 0.26 | 0.2 | 0.6 |
| 19 | 2 | 360 (5) | 355–365 | 347 | 0.26 | 0.2 | 0.5 |
| 20 | – | – | – | 348 | 0.26 | 0.1 | 0.3 |
| 21 | 1 | 420 | – | 348 | 0.26 | 0.1 | 0.2 |

**Table 4.** Results of paired *t*-tests of significant differences in length-at-age by geographic region [North Carolina through southeast Florida (NC-EFL) vs. The Florida Keys (FLK)], for individual ages of graysby (*Cephalophilis cruentata*) ageing samples collected 2001–2016. Comparisons were done for calendar ages for which there were *n* >10 samples from each region.

| Age | *n* (FLK) | *n* (NC-EFL) | Region |
|-----|-----------|--------------|--------|
| 4 | 32 | 81 | $t = -5.22, p < 0.0001$ |
| 5 | 25 | 96 | $t = -6.91, p < 0.0001$ |
| 6 | 37 | 143 | $t = -11.70, p < 0.0001$ |
| 7 | 42 | 138 | $t = -10.56, p < 0.0001$ |
| 8 | 35 | 116 | $t = -9.74, p < 0.0001$ |
| 9 | 46 | 95 | $t = -16.55, p < 0.0001$ |
| 10 | 33 | 74 | $t = -11.46, p < 0.0001$ |
| 11 | 23 | 79 | $t = -12.88, p < 0.0001$ |
| 12 | 13 | 52 | $t = -7.50, p < 0.0001$ |

After sub-setting the data by area, $n = 294$ individuals were from the southern (FLK) region and $n = 1014$ individuals were from the northern (NC-EFL) region. For the FLK region, posterior median parameter estimates (95% credible intervals) of the von Bertalanffy growth model were $L_\infty = 267$ (251, 299), $k = 0.17$ (0.09, 0.33), and $t_0 = -6.20$ (−2.17, −10.86). For the NC-EFL region, the estimates were $L_\infty = 388$ (369, 416), $k = 0.12$ (0.08, 0.15), and $t_0 = -5.73$ (−3.80, −8.29).

Because our data included few fish younger than age-3 (and no age-0 or age-1 fish), we do not believe the model was able to accurately capture initial growth for fish of ages 0–3, thus explaining the large negative estimates of $t_0$ (theoretical age at a length of zero). The lack of smaller fish is likely explained by gear selectivity, as our samples were all fishery-dependent. We re-estimated growth using a fixed value of $t_0 = -0.75$, which is consistent with spawning in the first quarter of the year as observed in graysby [12] and other grouper species [13]. The lower value of $t_0$ has the effect of pulling the growth curve down to simulate smaller fish length at age for the youngest ages (Figure 5; Residuals shown in Figure A3). With $t_0$ fixed, posterior median parameter estimates (95% credible intervals) were $L_\infty = 250$ (244, 257) and $k = 0.43$ (0.36, 0.55) for the FLK region, and $L_\infty = 349$ (344, 354) and $k = 0.26$ (0.25, 0.28) for the NC-EFL region.

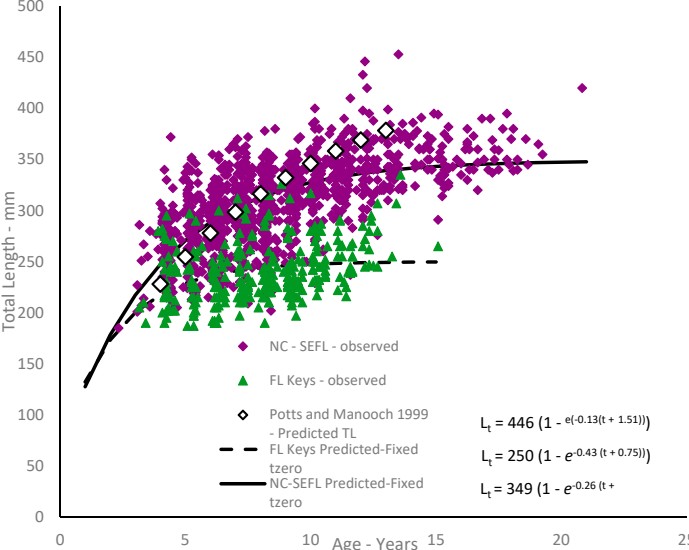

**Figure 5.** Comparison of observed size-at-fractional age of graysby (*Cephalophilis cruentata*) to von Bertalanffy growth curves for fish from North Carolina-southeast Florida vs. fish from the Florida Keys. Growth curves were estimated with a fixed $t_0 = -0.75$. The region-wide growth curve from the only previous published SEUS study [7] is shown for comparison.

### 2.3. Body–Size Relationships

Given the differences in growth by region, we analyzed the *W*–TL relationships by region as well. Statistical analyses revealed a multiplicative error term (variance increasing with size) in the residuals of the *W*–TL relationships for graysby, indicating a linearized ln-transform fit of the data was appropriate for both regions. The relationships are described by the following regressions:

FLK: $\ln(W) = -9.61 + 2.72 \ln(TL)$; $n = 349$. $r^2 = 0.87$; $p < 0.0001$, SE(a) = 0.13; SE(b) = 0.02;

and

NC - EFL: $\ln(W) = -11.73 + 3.12 \ln(TL)$; $n = 2113$, $r^2 = 0.89$;
$p < 0.0001$; SE(a) = 0.31; SE(b) = 0.05.

These equations were transformed back to the form $W = a(L)^b$ after adjusting the intercepts for log-transformation bias with the addition of one-half of the mean square error (MSE) [14], resulting in the following relationships (Figure 6):

LK: $W = 6.75e^{-5}\ TL^{2.72}$ (MSE = 0.021);

FNC − EFL: $W = 8.08e^{-6}\ TL^{3.12}$ (MSE = 0.027).

**Figure 6.** Comparison of region-specific TL-*W* regressions of graysby (*Cephalophilis cruentata*) measured by the Southeast Region Headboat Survey from the period 1975–2015 and a previous study consisting largely of northern region samples.

### 2.4. Natural Mortality

The age-invariant value [11] of natural mortality (*M*) was estimated to be 0.30 $y^{-1}$ for graysby, using the maximum age of 21 years from all samples. Age-specific estimates of *M* [10] are presented in Tables 2 and 3. For these values, we used the constrained growth estimates (i.e., with $t_0 = -0.75$).

Cumulative survivorship to each age (2+) based on the age-specific *M* [10] was similar between the two regions. Survivorship of graysby from the southern region ranged from 41% at age-2 to 0.5% surviving to age-15 (Table 2). Survivorship of graysby from the northern region ranged from 41% at age-2 to 0.5% at age-15 and 0.1% surviving to age-21 (Table 3).

Cumulative survivorship using the age-specific *M* was quite different from that using the age-invariant method. As expected, use of the age-invariant estimate of *M*, which was considerably less than the age-specific *M* for younger ages, resulted in a greater proportion of the population surviving

to each age. The survivorship estimates for graysby using the age-invariant method ranged from 74% at age-2 to 1.5% at age-15 and 0.2% at age-21.

## 3. Discussion

Otolith edge analysis demonstrated that graysby deposited one annulus per year from February–June, with peak annulus formation occurring in April. This agrees with a previous study that found a minimum marginal increment on the otolith edge in April [7]. These results are also similar to timing of annulus formation for other smaller members of the family Serranidae in the SEUS, which tend to form annuli in the late spring-summer months: coney—*Cephalophilis fulva* Linnaeus 1758, [15]; rock hind—*Epinephelus adscensionis* Osbeck 1765 [16].

Graysby attained an average observed size of 202 mm TL for Florida Keys fish and 239 mm TL for northern region fish by age-3. Northern region fish grew faster than southern fish, attaining average observed lengths by age-10 of 327 mm and 254 mm, respectively. This compares with a previous study [7] which reported an average observed size of 351 mm TL for graysby from the combined SEUS coast. While growth appears at first glance to be relatively slow (three years to achieve 200 mm), it should be noted that graysby attain about 75% of their $L_\infty$ by age-4 for northern fish and by age-2 for southern fish. While we may think of this species as slow-growing, the values of *K*, the von Bertalanffy growth coefficient, for each regional group indicate that they reach their theoretical maximum size at a moderate rate.

One limitation of this study is the lack of younger age classes, due to the fishery-dependent nature of our samples and the selectivity of fishing gear. This factor is likely the reason that we had a single fish younger than age-3. This lack of young fish common to studies dominated by fishery-dependent samples can lead to problems in estimating the initial trajectory of the growth curve for the youngest ages. This issue resulted in large negative values of $t_0$, which we took into account by re-estimating the growth parameters using a fixed value of $t_0 = -0.75$. This procedure had the effect of pulling down the initial trajectory of the growth curve (Figure 5), simulating a more realistic size-at-age for the youngest fish. Caution should be taken, however, when interpreting these estimated lengths-at-age beyond the range of the data, as any extrapolation may be unreliable.

Growth may also vary inter-annually in response to internal factors (e.g., density dependence) or external factors (e.g., environmental variables such as temperature, size-selective fishing pressure). Although we did not have enough samples to analyze whether there are inter-annual patterns in growth, this type of analysis would be valuable. It could be done in the future with increased sampling intensity (landings estimates indicate >10,000 fish landed annually on average).

Body–size relationships were different between regions in this study. The regression for the northern region was almost identical to the regression estimated for the previous study [7], and we speculate that this is due to the fact that 95% of their samples came from the northern region. We do not know exactly why fish from the southern region are smaller at a given age than northern fish. We hypothesize that environmental factors influence nutritional or energetic constraints. Such constraints underlie the trade-offs among growth, survival, and reproduction, and thus shape the life-history characteristics [17].

Natural mortality of wild populations of fish is difficult to measure but is an important component of stock assessments. For marine fishes in general, M is likely to be age-specific, decreasing as fish grow larger [10]. We have no reason to expect that graysby deviate from this pattern, but we also acknowledge that our age-specific estimates of M are uncertain, as they depend on growth curves that are themselves extrapolations at the youngest ages. By the age at which the fish are fully exploited in the fishery, age-7, the natural mortality rate at each age has generally leveled off and is close to the age-invariant estimate derived using maximum age [11]. Thus, either estimator would likely be suitable for the purpose of stock assessment [18].

When considering the cumulative estimate of survivorship to the oldest age, the age-invariant method [11] estimates 0.2% survivorship for the combined region. Estimates derived using the

age-specific method of calculating $M$ [10] are 0.1% and 0.5% for the northern fish (age-21) and southern fish (age-15), respectively. These estimates are supported by the age frequencies from both groups of fishes that indicate that the chance of survivorship to the oldest age may be less than 1%. These observations give weight to the argument to use the age-specific estimate of $M$ at age.

## 4. Materials and Methods

### 4.1. Age Determination and Timing of Opaque Zone Formation

Graysby samples ($n$ = 1318 fish) were collected dockside by NMFS and state agencies sampling landings from the recreational and commercial sectors along the SEUS coast during the period 2001–2016. All specimens were captured by either conventional vertical hook and line gear or divers with spears. Both fisheries sectors employed the same gear types, thus variable gear selectivity should not have influenced size of fish caught. All specimens were measured for total length (TL, mm). Additionally, fish landed by the recreational headboat fishery ($n$ = 2465) were weighed (whole weight, $W$, grams), and these weights were used in a W-TL regression analysis. Fish landed by commercial fisheries were eviscerated at sea, thus whole weights were not available.

Sagittal otoliths were removed during dockside sampling and stored dry. Otoliths were sectioned using a low-speed saw, two serial 0.5 mm sections were taken near the otolith core, mounted on microscope slides with thermal cement and covered with mounting medium before analysis [19]. The sections were viewed under a dissecting microscope at 12.5× using transmitted light. Each sample was assigned an opaque zone count by an experienced reader with extensive experience interpreting otolith sections [20,21]. Opaque zones were counted regardless even if they were not yet completely formed (i.e., no translucent zone beginning to form between the opaque zone and the edge). Sections were read with no knowledge of location or date of capture or fish size. After the initial reading, the otolith sections were set aside for several months and a subset ($n$ = 648) was re-read by the same reader. We calculated an index of within-reader average percent error IAPE [22]. Where readings for a specimen disagreed, the sections were viewed a third time and a final age determination was made.

Timing of opaque zone formation was assessed based on the distance between the outermost opaque zone and the edge of the otolith. Edge type descriptions are: 1 = opaque zone forming on edge of otolith section; 2 = narrow translucent zone of the edge, generally less than 30% of the width of the previous translucent zone; 3 = moderate translucent zone on the edge, generally 30%–60% of width of previous translucent zone; and 4 = wide translucent zone on the edge, generally greater than 60% of width of previous translucent edge [9]. The edge types were plotted by month of capture to determine if the opaque zones were deposited primarily in one season or month. Based upon edge frequency analysis, all samples were assigned a calendar age, obtained by increasing the opaque zone count by one if the fish was caught before that year's increment was formed and had an edge which was a moderate to wide translucent zone (type 3 or 4). Fish caught during the time of year of opaque zone formation with an edge type of 1 or 2, as well as fish caught after the time of opaque zone formation, were assigned a calendar age equivalent to the opaque zone count. Then, each fish was assigned a fractional age calculated from the calendar age, month of peak spawning and month of capture. Because we did not have a direct observation of the spawning season of graysby in our study area, we selected April as month of peak spawning based on observations on the species in the Caribbean [12] and other groupers [13]. Fractional ages were used for estimation of growth.

### 4.2. Growth

Growth was modeled for combined sexes due to the protogynous nature of graysby. We fitted the length-at-age (calendar age) data using the von Bertalanffy growth equation,

$$L_a = L_\infty\left(1 - e^{-k(a-t_0)}\right) \tag{1}$$

where $L_a$ is TL at age $a$, $L_\infty$ is the theoretical maximum length, $k$ is the Brody growth coefficient,

or the rate at which maximum size is attained, and $t_0$ is the theoretical age at size 0 [23]. We tested for whether the data were best explained by a single or multiple growth curves, using a new growth-morph approach [24]. The approach is described elsewhere in detail [24], and so we describe it here only briefly. In essence, it treats the growth data as a mixture of one or more components (growth morphs) from which an individual's identity may or may not be known a priori. We initially fitted the model allowing up to four morphs, with the identity (morph) of all individuals treated as unknown. We then considered three metrics in concert to indicate the optimal number of morphs: the deviance information criterion (DIC), leave-one-out cross validation (LOO), and the widely applicable information criterion (WAIC).

Based on results of the initial growth morph analysis, we re-fitted the model with identity treated as known, assigning identities based on geographic location (FLK or NC-EFL). We anticipated there would be few fish of the youngest age classes available to us, as our samples were primarily fishery-dependent, and hook-and-line gear or fishers generally selected for older or larger fish. Consequently, the model would be unable to depict initial growth of the youngest fish, leading to difficulty in accurately estimating size at the youngest ages. We therefore re-ran the growth model with $t_0$ fixed to a value of −0.75.

We assigned a uniform prior distribution to the asymptotic length of the largest growth morph, $L_\infty \sim U(300, 550)$. The asymptotic length of smaller morphs was computed as a proportion $c$ of the largest morph's $L_\infty$, similarly following a uniform prior distribution, $c \sim U(0.1, 0.9)$. We also assumed a uniform prior distribution for the growth coefficient, $k \sim U(0.01, 1.0)$, and a truncated normal distribution for $t_0$ with mean of 0.0 and precision (inverse variance) of 0.1. We used a truncated distribution to constrain values of $t_0$ to be negative.

The mixture model was fit in a Bayesian framework using JAGS version 4.3.0 [25], implemented in R version 3.4 [26]. The fitting procedure utilized three independent Markov chains, each of length 500,000 iterations. Posterior distributions excluded the first 100,000 iterations as the burn-in period, and we thinned by keeping every tenth iteration, because of data storage limits [27]. None of the posterior distributions were limited by the specified ranges of the prior distributions. We assessed convergence with visual inspection of trace, density, and autocorrelation plots, as well as with the Brooks–Gelman–Rubin statistic [28].

We conducted paired t-tests to determine if there were statistical differences in length-at age between geographic areas. For this analysis we used calendar ages, and only those ages for which we had adequate sample sizes ($n \geq 10$) from each area for comparison.

### 4.3. Body–Size Relationships

For weight–length relationships we regressed $W$ on TL ($n = 2465$) using all fish sampled for lengths and whole weights from the recreational headboat fishery during the period 1975–2015. We examined both a non-linear fit, $W = a\,\text{TL}^{\,b}$, using nonlinear least squares estimation [29] and a linearized fit of the log-transformed data, $\ln(W) = a + b\ln(\text{TL})$. Residuals were examined for patterns to determine which regression provided the best fit to the data.

### 4.4. Natural Mortality

We estimated the instantaneous rate of natural mortality ($M$) using two methods. The first method was based on maximum age of the whole sample, and resulted in an age-invariant estimate of $M$:

$$M = 4.889\, t_{max}^{\,-0.916} \tag{2}$$

where $t_{max}$ is the maximum age of the fish in the sample [11]. The second method was based on growth parameters:

$$M_a = (L_a/L_\infty)^{-1.5}\, k \tag{3}$$

where $M_a$ is natural mortality rate at age $a$, $L_\infty$ and $k$ are the von Bertalanffy growth equation parameters

and $L_a$ is fish length at age $a$ [10]. For these calculations, we used the midpoint of each age (i.e., 1.5, 2.5, 3.5, etc.), because the midpoint better represents the mean annual size.

The latter method, which incorporates life-history information via the growth parameters, is based upon evidence suggesting that $M$ decreases as a power function of body size. This method generates age-specific rates of $M$ as is commonly used in Southeast Data Assessment and Review (SEDAR) stock assessments e.g., [30].

Cumulative survival ($\phi$) to each age 2+ was calculated with both age-invariant and age-specific natural mortality rates:

$$\phi(a) = 100 \times \exp\left(-\sum_{a=1}^{A-1} M_a\right) \tag{4}$$

where $M_a$ is the natural mortality rate at age a, and $A$ is the age of interest (for age-invariant estimates, $M_a$ is constant). These survival rates represent expectation in the absence of fishing.

## 5. Conclusions

This study provides the most comprehensive analysis to date of graysby life history from SEUS waters. We have shown that otolith sections of graysby contain annuli that can be used for ageing. Opaque zones on graysby sagittae are assumed to be deposited once a year in late spring through summer. Our estimates of growth and natural mortality seem reasonable for a species with a moderate life span. While graysby are less commonly caught than many other species in the SEUS, understanding their life-history characteristics will support the push toward assessment of data-limited stocks and ecosystem-based fishery management.

**Author Contributions:** Conceptualization, M.L.B., A.D.O.; methodology, M.L.B., J.C.P., A.D.O, K.W.S.; software, J.C.P., K.W.S.; validation, M.L.B., J.C.P.; formal analysis, M.L.B., J.C.P., K.W.S.; investigation, J.C.P., A.D.O., K.W.S.; resources, J.C.P., A.D.O.; data curation, A.D.O., K.W.S.; writing—original draft preparation, M.L.B.; writing—review and editing, M.L.B., J.C.P., K.W.S., A.D.O.; visualization, M.L.B., J.C.P.; supervision, M.L.B.; project administration, M.L.B., J.C.P.; funding acquisition, J.C.P.

**Funding:** This research received no external funding.

**Acknowledgments:** We gratefully acknowledge the many NMFS headboat and commercial port samplers over the years whose efforts made this study possible. We thank N. Klibansky, K. Brennan, T. Kellison, and anonymous reviewers for comments that greatly improved the manuscript. The scientific results and conclusions, as well as any views and opinions expressed herein, are those of the authors and do not necessarily reflect those of any government agency.

**Conflicts of Interest:** The authors declare no conflict of interest.

## Appendix A

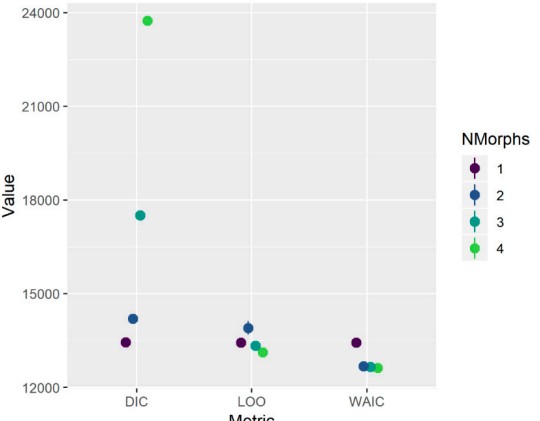

**Figure A1.** Information criteria for models with up to four morphs and individuals' identities (morph designation) treated as unknown. The metrics evaluated (lower is better) are deviance information criteria (DIC), leave-one-out cross validation (LOO), and widely applicable information criterion (WAIC).

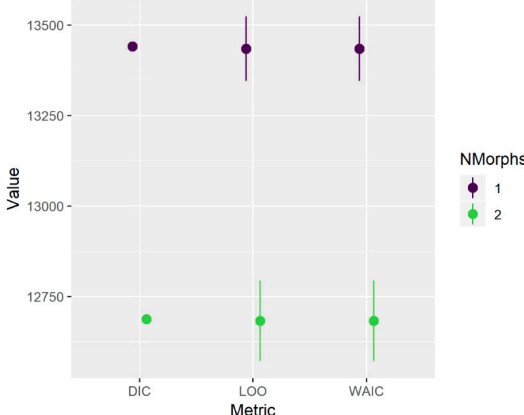

**Figure A2.** Information criteria (lower is better) for models with one or two morphs and individuals' identities (morph designation) treated as known. The metrics evaluated (lower is better) are deviance information criteria (DIC), leave-one-out cross validation (LOO), and widely applicable information criterion (WAIC).

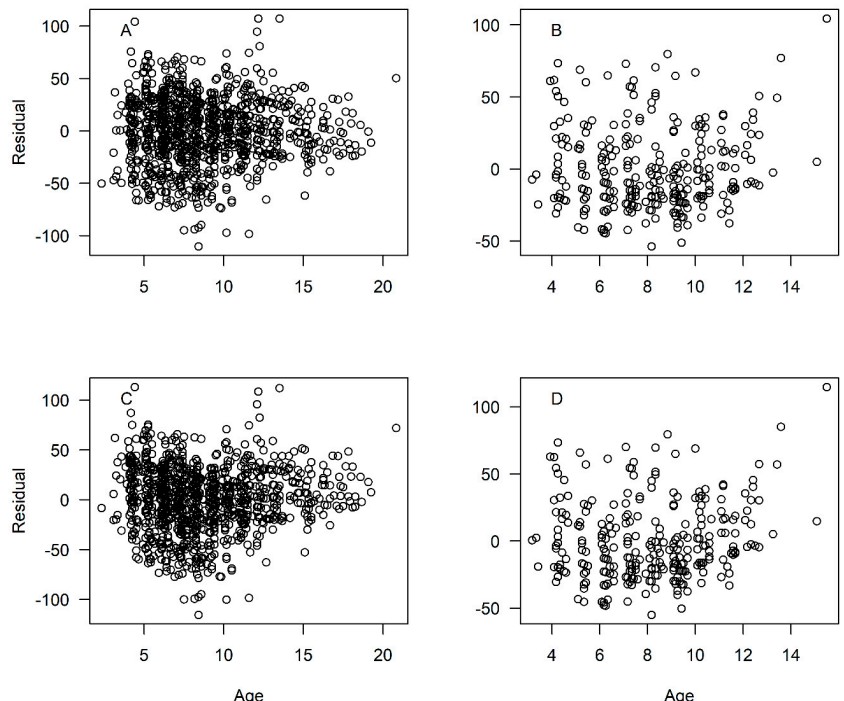

**Figure A3.** Residuals from model fits using median parameter estimates: (**A**) region NC-EFL with parameter $t_0$ freely estimated, (**B**) region FLK with parameter $t_0$ freely estimated, (**C**) region NC-EFL with parameter $t_0$ fixed (−0.75), and (**D**) region FLK with parameter $t_0$ fixed (−0.75).

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
