# Peer review of "Age, Growth, and Natural Mortality of Graysby, Cephalophilis cruentata, from the Southeastern United States"

_fishes, doi:10.3390/fishes4030036_

Round 1
Reviewer 1 Report
The work in the paper is useful and should be published.
I would like to see justification as to why the particular value of t0 of -0.75 was selected. The value of to has implications in the parameters estimates of the VB curve, from which the 'age-specific' estimates are dependent. If there is not a strong justification for -0.75, then perhaps the authors can give alternatives (higher and lower) and report whether this makes a marked difference to results.
Perhaps the authors should comment earlier in the manuscript about the impacts (or lack of) of sector selectivity on the resulting size at age - such that the reader has confidence the spatial differences are not due to sampling methodology.
Please include why the Potts and Manooch 1999 vb curve is included and its relevant to graysby.
Clarify which vb equations relates to which curve in figure 5, and complete the middle equation.
I agree with the authors that a larger sample size over time would allow analysis of variation in mortality over time and is a useful research recommendation.
Minor improvement
Table 2 - caption should note hat the t0 value was fixed at -0.75
Tables 2 and 3 - consistency in the table caption in regards to 'unweighted' VB parameters.
Table 4 - not sure if the n>10 rule was consistently applied and do you really in the ** if presenting the P values?
Author Response
I have attached our response to reviewers document, this addresses all comments by all three reviewers, i have included a line by line response.

Reviewer 2 Report
Review of paper by Burton et al entitled “Age, growth and natural mortality of graysby, Cephalophilis cruentata, from the southeastern United States
This paper provides a range of biological information of a small species of grouper in southeastern United States, relating to size and age compositions, and growth and mortality. These data are likely to be of local value for future stock assessments for this fish stock and thus, ultimately, for its management. A limitation of the paper is that, in its current form, it is descriptive and does not investigate hypotheses or issues likely to be of broad relevance to a wide readership. Further, there are a number of issues relating to analyses that need to be addressed. I’ll now turn to some specific points.
Lines 88-89: suggest clarifying, for cases when an opaque zone was visible (but not delineated) from the edge of an otolith, if that outer zone was included in the opaque zone count for that otolith.
Lines 104-108: outline in text what the various edge categories refer to (rather than just in the figure caption). No mention is made regarding the spawning period/estimated “mean” birth date for this species. If this is known, then a decimal age could be calculated for each fish. If not, then the true age is not known (only a relative age, i.e. the age relative to that when the first opaque zone is formed and delineated from the otolith edge), noting that good evidence is provided that the opaque zones are formed annually in this species.
Lines 128-131: Use of analysis of covariance does not appear appropriate for these data, as this test assumes linearity, but length-at age data are inherently non-linear. If the length-at-age data are assumed to be linear, then it would make sense to fit a linear regression to the data, rather than a non-linear curve such as the von Bertalanffy growth curve. The two alternative models (linear vs von Bertalanffy growth curve) could be fitted to the data for each region and compared to assess whether the more complex model (von Bertalanffy growth curve) provides a "better" fit to the data than a linear curve (e.g. AIC).
Line 169: No rationale is provided as to why it would be appropriate to set t0 to a negative value, and specifically, to -0.75 y. Why this value and not zero? t0 may be regarded as a ‘nuisance’ parameter and that in reality, fish cannot have a negative age (this only results from lack of representative data for young individuals). If a value is set for tzero, it needs to be clearly stated that the values of length at age estimated from the growth curve for fish below 2-3 y represents an extrapolation beyond the range of the data and are thus highly uncertain (and potentially unreliable). Note also that for stock assessment, many models can be modified such that knowledge of growth for very young fish is not required.
An alternative growth model that could be used is the reparametrized von Bertalanffy growth model described by Schnute 1981, in which lengths are estimated for two reference ages, together with the growth coefficient, k (both the hypothetical age at zero length, t0, and asymptotic length, Linf, can be derived from the estimated parameters, if desired). The growth curves for the different areas could be compared statistically, as could the estimated lengths for the two reference ages. This form of model would be more “robust” when fitting to data lacking small fish and should provide a good description of the lengths at age of fish in the sample.
Lines: 165-167. No model could capture the growth of the 0-3 year old fish well, as there are no data for such fish.
Growth curve figure: A residual analysis would be of benefit here. It appears that the growth curves may not be providing a good fit to the lengths at age of old fish when tzero is fixed (with these lengths being underestimated). Residual analysis will likely show a much better fit throughout much of the age range if t0 is not fixed (note that, due to selectivity, only the fasted growing individuals at the youngest ages are likely to be caught, which will also show on a residual plot).
Lines 196-199: What is the rationale for using different values for maximum age for estimating natural mortality, from the two regions? The more truncated age distribution from Florida Keys (and lower maximum age) may represent greater fishing pressure in that region, rather than higher natural mortality. Applying the same maximum age for both regions would appear more appropriate.
Lines 240-245. The differences in growth between the two regions, as the authors suggest, could relate to [evolutionary] life-history trade-offs. This is an aspect of the paper which could be developed, to increase its relevance to a wider readership. In particular, the authors could discuss their results in the context of the Metabolic Theory of Ecology (MTE).
Line 286: What is meant by ‘calendar age’? Suggest calculating a decimal age based on an estimated birth date.
Lines 304-308: The distributions for the priors for Linf and, more particularly, k, are quite precise, and thus have likely constrained the results. If so, the specified prior distributions do not appear appropriate.
Lines 328-330: As the growth coefficient, k, is poorly known (due to lack of data for small fish), use of this approach appears inappropriate. The more reliable method for estimating M would be that based on maximum age (of the whole stock).
Author Response
i have attached our response to reviewers document which addresses all comments by all three reviewers.

Reviewer 3 Report
This paper present a good study on the growth of Cephalophilis cruentata with a large dataset (n=1308) from 2001 to 2016. This format of this draft is very well.
i think that that draft could be considered for publication after major revision :
- statistical analysis : the authors present a good analysis to identify the best method per area but they did not presented some statistical analysis to compare the results between 2 areas as Kimura test for von bertalanffy model and analysis of covariance (ANCOVA) to length/weight relationship
- the authors used the subset to present opaque margin analysis, they had calibrated images and thus it will be better to use marginal increment analysis (MIA)
- the discussion is very short, in this area, there are the same trend of growth with others species... if yes why ? are there some difference between years ?
- there are 6 figures and 4 tables, it is possible to decrease this number with the figure 4 and Table 4 to sent to appendix
there are some other minor points :
- growth analysis without sex information because it is a hermaphrodite species, the authors could write this information in the introduction
- line 47 : no size limit > no commercial length
- Figure 1 : recreational or recreational but the same for both areas
- Table 1 : Totals> Total
- Figure 2 : this figure must be completed with growth axis and each growth ring for 2 images, perhaps only 1 image is enough. the right image is better than the left image
- Figure 3 : with MIA, the number per month and the statistical analysis showing the differeence among months with the same letter for exemple (anova).
- Figure 4 : it is not relevant in the publication, in the supplementary material
- Table 4 : it is not relevant in the publication, in the supplementary material
- Figure 5 : the second VB equation is not visible.. each VB equation must be with the sema color that observed data to understand, to add the number of data by each dataset.
- Figure 6 : to add observed data as Figure 5
- line 222, as the authors used Total length (TL), they must used only TL, no L in the text and the equation of VB... (line 287)
Author Response
I have attached our response to reviewers document, which addresses all comments by all reviewers.

Round 2
Reviewer 2 Report
The authors have addressed several of the issues that I had raised in my first review and the paper has been improved. Below, I discuss a few issues that I feel the authors should consider.
Calendar ages and specification of tzero at -0.75y
The authors have provided an explanation of their basis for setting tzero as a constant, i.e. -0.75y and also calculating “calendar ages” of fish, i.e. based on information regarding timing of reproduction (spring) relative to the beginning of the calendar year. From a biological viewpoint, I would imagine the authors would agree that the actual age of an individual fish in a sample must relate to i) mean time of spawning, ii) number of annually-formed zones in its otoliths, iii) time of year when zones are formed in the otoliths, and iv) the time of year when the fish was caught - but not to the beginning of the calendar year, Jan 1. The authors suggest that they have insufficient information to calculate a decimal age as reproductive data have not been published for this species from the same region, yet these data are being used to define “calendar ages” for all fish and set a value for tzero? From a biological viewpoint, I would consider it more justifiable to calculate a decimal age, recognizing that some uncertainty exists for the reason raised. The authors have also attempted to address my concern regarding their use of the term “calendar age” by adding the term “cohort age, which I also find confusing (as there are multiple cohorts present in the data – so which cohort is being referred to?). If the authors insist on using “calendar age”, then this should be added to the growth curve plot to indicate that the points do not represent true “ages”.
Prior distributions for growth parameters
In my previous review, I queried the prior distributions used for growth parameters, focusing in particular on the growth coefficient, k. The authors responded to this as follows “Prior distributions were initially selected so as not to be limiting of the posterior distributions. We had examined posterior distributions to ensure that this was the case, and we have added text to the methods to clarify this point.” There does still appear to be an issue, however, at least with the prior distribution specified for k, i.e. a uniform distribution between 0.01 and 0.3, noting that in the growth curve figure, for one of the areas, k is estimated as high as 0.47 – how can this be possible? I presume the actual prior distribution used was different to that presented, and that this needs correcting.
Estimate of M using an unreliable estimate for k
I can’t agree with the authors on this point. If there is insufficient information in the data to produce a reliable estimate of the growth coefficient (due to lack of curvature in the growth curve due to lack of small fish, or artificial curvature due to an imposed parameter value), then how can any approach calculating a variable derived from this k estimate be reliable? While I agree that, ideally, one would like to know the mortality rates for both younger and older fish, there appears to be insufficient information to do this, with the approach taken.
Author Response
Comments and Suggestions for Authors
The authors have addressed several of the issues that I had raised in my first review and the paper has been improved. Below, I discuss a few issues that I feel the authors should consider.
Calendar ages and specification of tzero at -0.75y
The authors have provided an explanation of their basis for setting tzero as a constant, i.e. -0.75y and also calculating “calendar ages” of fish, i.e. based on information regarding timing of reproduction (spring) relative to the beginning of the calendar year. From a biological viewpoint, I would imagine the authors would agree that the actual age of an individual fish in a sample must relate to i) mean time of spawning, ii) number of annually-formed zones in its otoliths, iii) time of year when zones are formed in the otoliths, and iv) the time of year when the fish was caught - but not to the beginning of the calendar year, Jan 1. The authors suggest that they have insufficient information to calculate a decimal age as reproductive data have not been published for this species from the same region, yet these data are being used to define “calendar ages” for all fish and set a value for tzero? From a biological viewpoint, I would consider it more justifiable to calculate a decimal age, recognizing that some uncertainty exists for the reason raised. The authors have also attempted to address my concern regarding their use of the term “calendar age” by adding the term “cohort age, which I also find confusing (as there are multiple cohorts present in the data – so which cohort is being referred to?). If the authors insist on using “calendar age”, then this should be added to the growth curve plot to indicate that the points do not represent true “ages”.
We have removed mention of cohort age. And we have re-fit the models using fractional ages, rather than calendar ages. This did not affect results much, but some. We have adopted this change into the paper, modifying all methods, results, appendices, tables, and figures as needed.
Prior distributions for growth parameters
In my previous review, I queried the prior distributions used for growth parameters, focusing in particular on the growth coefficient, k. The authors responded to this as follows “Prior distributions were initially selected so as not to be limiting of the posterior distributions. We had examined posterior distributions to ensure that this was the case, and we have added text to the methods to clarify this point.” There does still appear to be an issue, however, at least with the prior distribution specified for k, i.e. a uniform distribution between 0.01 and 0.3, noting that in the growth curve figure, for one of the areas, k is estimated as high as 0.47 – how can this be possible? I presume the actual prior distribution used was different to that presented, and that this needs correcting.
The reviewer is quite correct. Although we had initially used an upper bound of 0.3, we widened the interval for our final model runs. The upper bound was 1.0, and we have corrected this in the text. We thank the reviewer for catching this mistake.
Estimate of M using an unreliable estimate for k
I can’t agree with the authors on this point. If there is insufficient information in the data to produce a reliable estimate of the growth coefficient (due to lack of curvature in the growth curve due to lack of small fish, or artificial curvature due to an imposed parameter value), then how can any approach calculating a variable derived from this k estimate be reliable? While I agree that, ideally, one would like to know the mortality rates for both younger and older fish, there appears to be insufficient information to do this, with the approach taken.
We still believe that age-specific M is more biologically plausible for most marine fishes, however we also appreciate the reviewer’s perspective on the uncertainty surrounding our estimates. In response, we have rescinded our explicit recommendation that the age-specific estimator is best. Instead, we point out that age-specific M is preferred in general for marine fishes, but that in our case, these estimates are uncertain for the reason stated by the reviewer. We further point out, that for the purpose of stock assessment, this distinction between age-specific and age-invariant M might not be critical.
Reviewer 3 Report
Dear Authors,
the new version of this manuscript is fine to be publish.
Author Response
We thank the reviewer for his approval of our revised manuscript as submitted.